# “Freedom to Breathe”: Youth Participatory Action Research (YPAR) to Investigate Air Pollution Inequities in Richmond, CA

**DOI:** 10.3390/ijerph18020554

**Published:** 2021-01-11

**Authors:** James E. S. Nolan, Eric S. Coker, Bailey R. Ward, Yahna A. Williamson, Kim G. Harley

**Affiliations:** 1Center for Environmental Research and Children’s Health (CERCH), School of Public Health, University of California at Berkeley, Berkeley, CA 94704, USA; kharley@berkeley.edu; 2Department of Environmental and Global Health, College of Public Health and Health Professions, University of Florida, Gainesville, FL 32603, USA; eric.coker@phhp.ufl.edu; 3RYSE Youth Justice Center, Richmond, CA 94805, USA; baileyrward@gmail.com (B.R.W.); yahna234y.w@gmail.com (Y.A.W.)

**Keywords:** youth empowerment, air pollution, inequity, structural violence

## Abstract

Air pollution is a major contributor to human morbidity and mortality, potentially exacerbated by COVID-19, and a threat to planetary health. Participatory research, with a structural violence framework, illuminates exposure inequities and refines mitigation strategies. Home to profitable oil and shipping industries, several census tracts in Richmond, CA are among the most heavily impacted by aggregate burdens statewide. Formally trained researchers from the Center for Environmental Research and Children’s Health (CERCH) partnered with the RYSE youth justice center to conduct youth participatory action research on air quality justice. Staff engaged five youth researchers in: (1) collaborative research using a network of passive air monitors to quantify neighborhood disparities in nitrogen dioxide (NO_2_) and sulfur dioxide (SO_2_), noise pollution and community risk factors; (2) training in environmental health literacy and professional development; and (3) interpretation of findings, community outreach and advocacy. Inequities in ambient NO_2_, but not SO_2_, were observed. Census tracts with higher Black populations had the highest NO_2_. Proximity to railroads and major roadways were associated with higher NO_2_. Greenspace was associated with lower NO_2_, suggesting investment may be conducive to improved air quality, among many additional benefits. Youth improved in measures of empowerment, and advanced community education via workshops, Photovoice, video, and ”zines”.

## 1. Introduction

The field of public health has facilitated major gains in improving societal level health, especially over the last 100 years. Many of these improvements have been broad strokes interventions, such as removing lead from gasoline and paint, improvements in occupational health and safety, improving water quality and ensuring food safety, affecting the majority of people in the US [1,2]. Yet, health disparities persist [3,4], and the field is increasingly faced with complex challenges of historically rooted structural determinants of health. These challenges include a lack of socio-historical context, misalignment between community and researcher goals [5], insufficient data granularity and interventions that address specific downstream symptoms rather than root causes of problems [4]. Scientific-grade youth participatory action research (YPAR) studies can help address this gap by leveraging youth’s experiential expertise to fine tune scientific data collection protocols for context-specific scenarios, building relationships between researchers and community, collecting nuanced neighborhood level data responsive to local needs [6], framing findings in local vernacular and contexts, and disseminating results more effectively. In recruiting youth from impacted communities and providing them with mentorship, afterschool educational opportunities, skills development, paid work, college preparation and amplifying their voices, YPAR can help dismantle long-standing power dynamics [7]. The model prioritizes centering impacted communities, building local capacity and expanding future opportunities for improved wellbeing [8,9,10].

From 2017 to 2018, the Richmond Youth Air Quality Initiative utilized YPAR to investigate air quality and related environmental justice challenges across neighborhoods facing heightened adversity in Richmond, CA, an industrial community in the San Francisco Bay Area with high disadvantage and pollution that is situated on Ohlone land [11]. Researchers from UC Berkeley’s Center for Environmental Research and Children’s Health (CERCH) partnered with RYSE, a community youth center in Richmond promoting social justice and development opportunities. The project facilitated empowerment among high school youth through: (1) training in environmental health literacy (EHL), professional development, paid employment and college preparation; (2) full partnership in designing and conducting research on the convergence of structural air pollution and community-level health risk factors using both quantitative and qualitative methods; and (3) interpretation of findings, community outreach and advocacy. Youth were facilitated in exploring manifestations of structural violence, explicitly examining features in the built environment that may affect pollution levels in correlation with pre-existing socio-economic burdens, monitoring for disparities in NO_2_, SO_2_ and noise pollution and returning results to the community.

## 2. Background

### 2.1. Air Pollution

Inequitable distribution of air pollution is a major driver of health disparities. Globally, it is estimated that air pollution contributes to seven million preventable deaths each year, largely from respiratory infection, chronic obstructive pulmonary diseases, heart disease, and cancer [12]. Exposure during pregnancy is associated with low birthweight, premature birth and infant mortality [13]. The vast majority of those most impacted are low-income people of color [12]. Air pollution also presents a significant threat to planetary health, with emissions exceeding earth’s absorption rate, risking global warming, mass extinctions and ecological collapse. The health effects of air pollutants among sub-groups with heightened biological susceptibility and vulnerability remains understudied. Susceptible sub-groups include the very young, the very old, those with respiratory diseases and those who face high levels of socio-economic disadvantage and other chronic social stressors. Emerging findings indicate that air pollution may exacerbate both the incidence and harm of COVID-19 [14,15], a disease that disproportionately affects communities of color [16,17].

Air pollution is classified as outdoor (ambient) or indoor and includes a range of sources and exposure determinants. Concerning monitoring and human health effects, the most widely studied air pollutants include particulate matter (PM), ozone (O_3_), nitrogen dioxide (NO_2_), and sulfur dioxide (SO_2_) [18]. NO_2_ can contribute to PM2.5 and ozone air pollution respectively, and is associated with increased risk of pulmonary distress and asthma symptom exacerbation [18]. SO_2_ is the leading cause of acid rain, which is devastating to ecological balance and is associated with respiratory distress, exacerbation of respiratory disease symptoms, increases in overall mortality and cardiac disease hospitalizations in humans [18]. Emissions generating activities are largely driven by commodity production, transportation and participation in growth-focused global markets, and include stationary sources such as heavy production industries and mobile sources such as trucks and trains. Though mobile sources are transient by nature, they generally follow well-established routes, resulting in clusters of heightened emissions.

Our research focused on anthropogenic air pollutants, which are readily measurable and more amenable to mitigation than natural sources such as forest fires, which themselves may also be increasing in intensity and frequency due to increasing air pollution and subsequent global warming [19]. Specifically, we measured urban ambient NO_2_ and SO_2_ concentrations and studied spatial variability of pollutant concentration within Richmond, CA. Selection of NO_2_ and SO_2_ as study pollutants were determined by the following factors: (1) a priori relationships with localized traffic and industry-related emissions sources; (2) potential for human health impacts; (3) potential for spatial heterogeneity in concentrations; (4) potential as a target for future pollutant mitigation efforts; and (5) ability to be reliably and practicably measured and interpreted using low-cost sampling devices. Spot noise pollution data was also collected, as some emerging evidence has suggested a negative relationship with several health outcomes [20,21].

### 2.2. Structural Violence

#### 2.2.1. Framework

Contextualizing neighborhood level spatial distributions of air pollution data using a structural violence framework helps illuminate the intrinsically meaningful context within which pollutant emissions and exposures occur. Structural violence is defined as “the prevention of individuals or populations from reaching their mental and physical potential without a specific actor committing the violence; when ’violence is built into the structure and shows up as unequal power and consequently as unequal life chances.’ It may be unintended and indirect [22]”. Structural violence analysis unveils clustering of environmental harms, such as the inequitable proportion of hazardous sites in disadvantaged communities [3]. Implications support centering populations most affected, and social justice driven pathways for remediation. Here, justice may imply not just the cessation of harm, but also recompense for harms already done.

#### 2.2.2. Transgenerational Structural Violence

Histories of structural violence precede, and persist in, contemporary injustices, thus transgenerational violence must be integrated into analysis and intervention. Historical violence against marginalized communities sculpt “*colonial geographies of harm*” [23]. Value-laden borders based on social identity manifest in physical space and morph in response to forces and resistance, fueled by such concepts as “The West”, colonialism and modernity [23]. These concepts become embedded in contemporary structures, ideological made physical, and normalized over long periods [24], making them difficult to identify and even more so to act against. Examples include Western colonist’s *attempted* genocide against Native Americans and forced relocation to reservations on “marginal” land [24], Trans-Atlantic slavery, and racial segregation in the early 20th century. In each instance, extreme force was used to establish boundaries, cordoning low income and people of color, and policing behaviors. In the face of resistance, explicit force transitioned to more insidious mechanisms such as withheld GI-Bill housing and education support for Black veterans of WWII [25], economic divestment and race-based exclusion via redlining [26], racially biased federal loan practices favoring white farmers [27], and the mass incarceration of people of color [28]. Here, the opportunities that groups of people are held back from, and the spaces they are held in, engrained in replicating and obfuscated structures, become violence.

#### 2.2.3. Contemporary Structural Violence

Insidious constellations of inequity make it appear as though individuals, not systems of socio-historical violence, are at fault [29]. Culturally structured white supremacy, coded with alleged racial identifiers and thus transgenerational, shifted to wealth passed on in such forms as company ownership, stocks and estate inheritance [30,31], as well as status, privilege and opportunity access, themselves passed on transgenerationally [32,33]. Neoliberalism accelerated wealth accumulation and eroded social services [34], while fossil fuels accelerated neoliberalism, and air pollution, on a global scale. In 2019, half of the world’s top 10 corporations are fossil fuel companies [35], and just three white men control more wealth than the bottom 50% of US residents combined [36]. Of the 100 wealthiest people in the US in 2018, 90% were male, the vast majority were white and only 1 was Black [37]. In 2019, the average white family had roughly 8 times the wealth of the median Black family, and 5 times that of a Latinx family with little improvement over the last several years [38]. Those who experience poverty for 50% or more of their childhood have a 35–46% chance of being in poverty themselves as they become middle aged [39]. Income inequality itself can be a significant cause of health disparities [40].

Inequity tracks with exposure burdens and resultant health impacts along strata of race, place and class [41,42,43,44]. For example, Blacks in the US face roughly 50% more air pollution relative to their consumption, while whites face 17% less than theirs [45]. Lauren Berlant’s concept of “slow death”—“the physical wearing out or deterioration of people that is very nearly a defining condition of their experience and historical existence [46]”—speaks to an amalgam of socio-economic factors that compound such that a Black child born in a disadvantaged Oakland neighborhood is expected to have a 15 year shorter life compared to a white child born in the hills of that same city [47]. Oakland borders Berkeley, where this paper was originally written, with Berkeley just a few miles from Richmond, about which this paper is written.

#### 2.2.4. Contemporary Resistance to Symptoms of Structural Violence

Millions in the US have been catalyzed by mass social movements in addressing symptoms of structural violence. Over the last decade, Occupy/Decolonize Wallstreet, #MeToo and Black Lives Matter have focused on mitigating harm in relation to economic disadvantage, gender discrimination and racism respectively. In 2020, George Floyd—raised in a low-income, single-mother household, and a Black man—was killed by police in an egregious act of violence [48]. Though complex, research indicates a strong connection between being low income (male) and Black, and higher probability of police arrest, imprisonment [49], and police violence, with police force being a leading cause of death for young Black men [50]. Re-energized by the pursuit of justice for Floyd, as well as for many before and after him, the Black Lives Matter movement has become the largest US social movement to date [51]. The movement is rooted in resistance throughout deep histories of slavery and continuing racial injustice. Many such injustices have been overt and acute for 400 years. With advances in video recording and communications technology, contemporary manifestations of these historically rooted injustices can now be transmitted, and better articulated, almost instantaneously across vast networks, perhaps therefore increasing societal level engagement. The “slow death” of oftentimes invisible, gradual, and mechanistically complex air pollution exposures, frequently the result of unintended and indirect forces, oftentimes manifest in a holding back and a holding within marginalized “*geographies of harm*”, stand in contrast. Aptly stated by Foucault, in a distinct but complementary context “... invisibility is a guarantee of order [52].” Noteworthy also, is that George Floyd, deprived of his “freedom to breathe [53]” by state violence, was also diagnosed with COVID-19 [48], itself an airborne respiratory disease disproportionately burdening low income and communities of color in the US, and potentially exacerbated by air pollution. Implications, including potential synergies between increasing public awareness around Black Lives Matter, disparities in the effects of COVID-19, and improved environmental justice are overviewed in the Discussion section.

#### 2.2.5. Structural Violence in Richmond, CA

Our study was conducted in Richmond, land historically occupied by Ohlone Native Americans and currently occupied by diverse communities of color, including members of the Ohlone. In early to mid-1900s the city had a booming economy and included a high proportion of white residents. As those industries, many based on a war effort, became less profitable, many white residents moved out [54]. Today, 20.6% of residents identify as Black, and 42% Latinx [55]. As many as 20% of Richmond’s residents are undocumented [56]. The City of Richmond has made significant gains in improving housing stock, reducing crime [57] and establishing a Health in All Policies campaign for addressing its multifaceted community health challenges [58] and several personnel have taken notable steps to support this study, as well as others like it. An array of organizations are confronting environmental injustice, marginalization of youth, and community violence.

Despite progress, Richmond remains an at-risk community due to structural inequities, including high emissions sources and population-level vulnerabilities [59,60]. Roughly 17,000 people reside in the neighborhoods of Atchison Village and North Richmond, census tracts ranking in the top 90–95th percentile statewide for combined socio-economic and pollution burden [60]. The population in these tracts is >92% non-white, with asthma burdens in the 99–100th percentiles, and diesel burdens in the 84–95th [60]. Richmond houses the largest oil refinery west of the Mississippi, the cause of several air pollution emergencies in recent years including one that hospitalized 15,000 residents [61], resulting in great concern among many in the community. Previous community-based participatory research revealed that air toxics concentrations, as well as the variety of air toxics, were substantially higher in Richmond than a neighboring, predominantly white and more formally educated community [62]. Richmond faces a violent crime rate almost twice as high as the surrounding county and state as a whole [63]. While 15% of those over 25 years of age in the US do not have a high school diploma, this proportion is 32% or higher in impacted Richmond census tracts, as high as 58% [64]. Physical activity and access to healthy foods are major challenges as well, with nearly half of the population obese or overweight and nearly a third living in food deserts [65].

### 2.3. Air Quality Research Gaps in Richmond

Despite regulatory air monitoring networks throughout U.S. cities, major urban air quality data gaps exist. Because the primary goal of regulatory monitoring is to assess whether a region’s overall pollution levels fall within air quality standards, these networks are ill-suited to examine spatial heterogeneity of air pollution levels *within* cities. Though several of Richmond’s neighborhoods are impacted by emissions from heavy industry and multiple highways, the city has just one regulatory air monitoring site that measures criteria air pollutants regulated by the US EPA (e.g., NO_2_, SO_2_, PM2.5, ozone, and carbon monoxide). Consequently, little is known about differences in air pollution levels between Richmond neighborhoods. Such limitations leave densely populated areas with potentially insufficient air quality data, which could otherwise guide policymakers in prioritizing local air pollution mitigation efforts. The spatially dense network of air samplers implemented in our study enabled youth researchers to better characterize spatial variability of air pollutant concentrations between Richmond neighborhoods and associate this variability with features of the built environment, many of which exemplify continuing structural violence.

### 2.4. Youth Participatory Action Research (YPAR) Model

As nuanced socio-economic factors may affect the likelihood that persons of certain groups, especially communities of color, are disproportionately exposed to environmental health challenges such as pesticides [66], poor air quality [67], or impaired drinking water [68], and the severity of negative health outcomes, it is important to include members of impacted communities in the research process. Through lifetimes of experience in affected communities, locals gain invaluable insight into mechanisms behind key challenges, intersectional harms and assets to address them. This process of engagement frequently reveals unforeseen synergies between organizations, community members and researchers. The “gold standard” model for community researcher partnerships is Community Based Participatory Research (CBPR). CBPR emphasizes strengths-based engagement, equitable community inclusion in all facets of research, co-learning between academics and community members, equitable distribution of resources, local capacity building, research and action, addressing local priorities, and sustained engagement [69]. CBPR improves “relevance, rigor and reach” [70]. Community input guides study foci and ensures *relevance* to community needs. *Rigor* is improved as residents have experiential expertise and can highlight contexts affecting local exposure or vulnerability. Locals may be perceived as more trustworthy than outsider researchers and granted greater access to sensitive information or culturally significant locations. CBPR enhances the *reach* of findings, as locals know how to frame results using appropriate language that appeals to those most affected, and ideal venues for dissemination [70].

This study used YPAR, a corollary of CBPR with emphasis on youth empowerment and trajectories of development. There is a great body of research indicating that YPAR studies can enhance research, and outcomes for youth themselves [71,72,73,74,75,76]. Some stipulate that pre-adult years constitute a “sensitive period” where experiences can make disproportionately positive effects on overall trajectories of identity and civic engagement [69]. Yet, as they are still learning, youth may be somewhat less able to inform initial study design, and somewhat less able to engage in advanced facets of data analysis. In balance, youth bring a fresh perspective and are more prone to unorthodox and real-time problem solving, often necessary given limited resources.

## 3. Methods

### 3.1. YPAR Methods

CERCH engaged youth in harnessing scientific grade technologies for granular quantitative data collection, facilitated youth in enriching this data with qualitative data, compensated youth for their time and, on request, provided supplementary skill building workshops. Detailed in previous publications, the underlying framework, sequence, and rationale of CERCH’s YPAR approach is summarized elsewhere [77,78]. CERCH staff, in collaboration with RYSE and with input from Bay Area Air Quality Management District experts, created a YPAR project and accompanying environmental health literacy curriculum to engage Richmond youth as co-researchers. Curriculum drew from years of researcher experience and field-tested activities publicly available on UC Berkeley’s YPAR Hub [79] as well as from nearly 10 years of CERCH YPAR expertise. Youth researchers were provided a ~150-page reader assembled specifically for this project by staff, which included self-care tips, background readings, tox-fact sheets, relevant research, conceptual frameworks, case studies and potential intervention ideas.

Meeting content and study design progressed similar to previous CERCH YPAR studies. Sequentially, youth built group trust, learned key foundational concepts and gained increasing control over study activities, engaging in roughly 100+ hours of meetings and activities and dozens of hours working independently. Meetings were held at RYSE, a location convenient for the youth and where an array of additional youth opportunities were available free of charge, including arts, education and justice, organizing, free meals, and community health. Youth drew upon their personal experiences with air quality challenges in their daily lives, identifying potential sources of emissions, and brainstorming risk and protective factors in the vicinity. Youth were paid stipends for their engagement. Local environmental justice leaders with expertise in air quality, the vast majority of whom were people of color (and offered an honorarium), informed the youth about key achievements, lessons learned and community concerns. Speakers included representatives of CERCH, the Bay Area Air Quality Management District, Greenaction, West Oakland Environmental Indicators Project, Air Watch Bay Area and Communities for a Better Environment. Youth did background research on each speaker to shape the discussion, creating prompts to acquire information relevant to the study.

CERCH researchers with expertise in air quality research trained youth in the use of scientific-grade air samplers, including techniques to collect, analyze and interpret air quality data. Youth used CalEnviroScreen Community Disadvantage maps and, drawing upon their experiential expertise in these specific neighborhoods, brainstormed key locations where vulnerable populations may spend time, such as community centers, hospitals, schools, stores and transit hubs. Visualized through these tools, youth discussed known factors related to community vulnerability, air quality and other environmental health threats that could compound the harms of poor air quality, and identified gaps that could be filled by improved data collection in these spaces. A selection of 12 sites from this larger pool of options were voted upon, balancing the potential significance of the site with need for sufficient distance between each sampler to ensure adequate coverage within the target census tracts. Prior to sampler deployment, study personnel and the youth visited the sites to determine whether they were suitable for sampler installation. Considerations included whether there were available streetlight poles to attach samplers to, whether the terrain was safe to use a ladder on for deployment and other physical impediments. Minor adjustments were made accordingly. A combination of Microsoft Excel, Google Spreadsheets, Google Docs, Microsoft Powerpoint, Adobe Photoshop and Google Fusion Tables were used to explore and co-analyze the data with the youth.

The project’s effects on the youth council members was gauged using pre- and post-test questionnaires that included validated scales [77] coupled with qualitative feedback via answers to reflection questions. To date, empowerment evaluation tools used for this study have been used in several YPAR studies conducted by CERCH: Leadership Efficacy, Resource Mobilization and Leadership Behavior, Community Engagement, Action Self Efficacy and Socio-Political Skills [80]; Motivation to Control [81] Science Attitude, Self Esteem and Motivation to Influence; and Participatory Behavior [82].

Data was collected and managed according to a protocol approved by the University of California Berkeley’s Committee for the Protection of Human Subjects (protocol No. 2017-04-9899), by CITI certified staff trained in public health. Youth empowerment data collection activities were detailed on approved assent forms (for minors under 18 years of age), or consent forms for those over 18, which were verbally outlined to youth. It was clarified that participation in evaluation was completely optional and that youth’s participation, or lack or participation, in evaluation activities would not affect youth’s ability to participate in the project overall or to access related opportunities. Youth under 18 years of age acquired parent signatures on a similar consent form, without which no data would be collected.

### 3.2. Air Quality Data Collection Methods

With extensive input from the five youth researchers, a cohort limited in size by funding constraints, our research targeted Richmond census tracts most impacted by environmental and social burdens. We concentrated sampling efforts in two of the most socio-economically burdened tracts discussed earlier, due to high population density and the presence of sensitive subpopulations (e.g., young children, those with asthma) as identified by the youth researchers. Our project combined scientific methods with community engagement to: (1) better understand spatial features related to between-neighborhood differences in outdoor air quality; (2) to investigate whether spatial variability in pollutant levels contributes to potential population exposure disparities; and (3) to better understand how spatial variability in multiple outdoor air pollutants correspond with Richmond’s community-level risk factors. Passive samplers were selected as they are reliable for gaseous air pollution measurement and well suited for community-involved research.

Youth researchers selected 12 air sampling sites based on knowledge gained from curriculum on air pollution sources and susceptible sub-populations. Before entering the field, youth were quizzed on the functions of key materials, and their use. Youth engaged in adapting a protocol and practiced deployment on a pilot day. Checklists with images were created to efficiently identify materials, follow the appropriate order of operations and inventory supplies. Checklists allowed youth to easily double check equipment and each other, building internal cohesion and teamwork skills. With supervision, youth deployed 12 Ogawa passive samplers in residential neighborhoods in Atchison Village, North Richmond, and areas south of Atchison in proximity to I-580. Samplers were placed at each site for three weeks, from mid/late-December to mid-January (see Figure 1. Spot noise measurements were performed (~10–15 min per sample site) during deployment and retrieval. Data on community-level risk factors in sampled communities were collected and analyzed for cumulative neighborhood vulnerabilities. Youth took the lead on Community Mapping and PhotoVoice exercises, which helped to further contextualize combined neighborhood risk factors.

### 3.3. Data Analysis

Sample sites were mapped (Figure 1) and statistical and spatial analyses of air quality and noise data were performed. Analyses were informed based on research questions from youth researchers. Together, we determined the following air quality and emissions source indicators: neighborhood differences in NO_2_ and SO_2_ concentrations; major roadway density for 250 and 500 m circular buffers around each sampling site; roadway density and pollutant concentrations around sampling sites; railway density in 250 and 500 m circular buffers; greenspace (normalized differential vegetation index (NDVI)) in 250 and 500 m buffers; and federally regulated air emissions site density in 250 and 500 m buffers. To assess possible differences in cumulative impacts of air and noise pollution measurements, and emissions sources, summary scores were computed for the different indices. For example, the summary score for emissions sources was computed by first computing a Z-score for the density of each type of emission source indicator and summing these values for each sampling site. Univariable and multivariable linear regressions were used to determine the relationships between emission source indicators (e.g., roadway density within 250 m) and air pollutant concentrations.

## 4. Results

### 4.1. Air Quality Results

A summary of measured NO_2_ and SO_2_ concentrations and noise levels are in Table 1. A detailed summary of pollution levels for each site is in the Appendix A. For context, during the same timeframe of our study, the average NO_2_ concentration measured in our study (12.09 ppb) is similar to the average measurements made at the nearest U.S. EPA monitoring station (13.24 ppb) that is located in the neighboring City of San Pablo. For SO_2_, however, measurements made in our study were not similar to the nearest US EPA ambient SO_2_ monitoring station (see Table 1). Figure 1 displays spatial locations of each of the 12 air sampling sites. Significant differences in neighborhood ambient NO_2_ concentrations were observed (Figure 2), with a significantly lower level of NO_2_ observed for sites in North Richmond. Linear regression models indicated higher levels of greenspace (NDVI) were significantly associated with lower concentrations of NO_2_ (Appendix A). Greenspace within 250–500 m of sites explained 73% (*p*-value < 0.001) and 71% (*p*-value < 0.001) of NO_2_ variability, respectively. The spatial density of railroads within 250 m (R^2^ = 0.45; *p*-value = 0.02) and major roadway density within 500 m (R^2^ = 0.32; *p*-value = 0.055) were associated with higher NO_2_ concentrations. Noise levels were also positively associated with NO_2_ concentrations (R^2^ = 0.30, *p*-value = 0.08). While aggregate NO_2_ concentrations showed little correlation with the density of industrial emissions sites, the site with the highest concentrations of NO_2_ (14.9 ppb) was closest to a major oil refinery and was 12% higher than the NO_2_ measurements from the nearest U.S. EPA ambient air quality monitoring station in San Pablo. Concentrations of SO_2_ did not show significant associations with spatial features, with noise levels, or between neighborhoods. As suggested in Figure 3, census tracts with the highest Non-Hispanic Black populations (south of Atchison) experienced the highest NO_2_ concentrations, the highest levels of cumulative air and noise pollution levels, and the highest cumulative density of emissions sources. In addition, tracts south of Atchison with the highest proportion of non-Hispanic Blacks had the lowest levels of neighborhood greenness (Appendix A). Refer to the Supplement (Appendix A) for further spatial distributions of densities and built environment features considered. We note that although we conducted only spot measurements of sound levels at our monitoring sites, the average site decibel levels recorded in our study exceeded the City of Richmond’s noise ordinance limits for exterior noise for every zoning type except for Heavy and Marine Industrial Zones (the highest zoned area for noise). The average noise measurement detected in our study reached the Heavy and Marine Industrial Zone ordinance threshold of 75 decibels [83].

### 4.2. YPAR Results

Results are drawn from a very small sample size, *n* = 5, limited due to funding constraints. Though there were no exclusion criteria for participation other than being of high school age and residing in the Richmond area, no youth who remained committed throughout the entire project year identified as white, and all identified as female. Similar to previous CERCH YPAR studies, limited resources meant that extensive qualitative data could not be collected. However, results parallel those of previous CERCH YPAR evaluations that had greater sample sizes. Of 68 total fields, and despite relatively high baseline measurements in many fields, results improved in nearly every category. The 11 most notable improvements appeared to be patterned in fields related to youth’s community engagement, attitudes towards one’s neighborhood and voicing one’s opinions (Table 2).

Personal responses to qualitative reflection prompts indicated youth’s successful acquisition of knowledge related to key concepts, that the project was perceived as impactful and that youth improved in environmental health literacy. Youth’s successful completion of data collection and peer education activities is also indicative of project success, as is their library of ~1500 high resolution photographs of assets and challenges in the community, their “zine”, a short video they made about youth’s engagement in science and several successful presentations. Anecdotally, staff observed an exceptionally high degree of team building, and an overwhelmingly positive and collegial atmosphere between youth themselves and with staff during meetings and project tasks.

Beyond the scope of the project, steps were taken to ensure that youth’s needs were met, supporting their access to opportunities. As youth were applying to college, extra workshops were offered on college application personal statement writing, key dates, and application tips. Youth were given the option of submitting their statements to staff for review and provided with extensive feedback. Youth were also invited to tour UC Berkeley and attended a panel with UC Berkeley undergraduate students of color who spoke about their experiences and provided tips. All youth were accepted to, and attended, leading universities. Verbally, 60% expressed that because of this program they would major in environmental studies or environmental science. The remaining 40% expressed they planned to integrate lessons learned about environmental health into their fields of study. 100% applied to external summer employment opportunities that related to environmental health and 40% were able to secure paid employment, directly addressing air quality concerns. To date, staff have been able to support 40% of youth in obtaining additional paid placements with environmental health organizations and 60% in presenting to graduate level classes. Though these were not explicit study objectives, they reflect the spirit of participatory research, where researchers take measures to support capacity and facilitate opportunities to access important resources. We recommend that researchers engaging in participatory research take similar measures to ensure organic and iterative support beyond explicit study aims, especially those with the privileges associated with being part of an academic institution, with advanced credentials or with otherwise high socio-economic status.

## 5. Discussion

### 5.1. Air Quality

Findings show that multiple features of the built environment were significantly associated with NO_2_ concentrations. These correlations were in the expected directions (e.g., railways were positively correlated with NO_2_). Possible cumulative impacts were also observed in our study. Noise levels were correlated with higher levels of NO_2_, suggesting that communities experiencing higher NO_2_ levels are doubly burdened with noise pollution. Sampled census tracts with the highest proportion of Non-Hispanic Blacks experienced the highest levels of NO_2_. Again, this suggests that Non-Hispanic Blacks, a historically marginalized group, may be doubly exposed to higher levels of air pollution and social stressors compared to other Richmond communities monitored in our study. Greenspace was significantly associated with lower NO_2_ concentrations and was also the strongest predictor of NO_2_. This suggests that greenspace may mitigate some air quality concerns. Moreover, greenspace has been shown in other studies to be conducive to improvements in physical activity [84,85] and real estate value [86,87], thus investment may address multiple symptoms of structural violence. Research further indicates that greenspace may have positive impacts on a range of human health factors [88,89]. A final report was submitted to the funder, the Bay Area Air Quality Management District. When published, this paper will also be submitted to BAAQMD, as well as personnel at the California Air Resources Board, the National Institutes of Environmental Health Sciences and additional colleagues.

Evidence indicates that remedial air quality policies backed by enforcement can yield high socio-economic gains. For example, while it is estimated that actions needed to meet US Clean Air Act benchmarks would require USD 65 billion from 1990 to 2020, the positive economic value of air improvements over those 30 years was projected to be USD 2 trillion, a return of more than 30 to 1 [90]. It has been estimated that the economic value of eliminating health disparities more broadly is in the trillions of dollars [91] with savings from eliminating racial health disparities specifically estimated in the hundreds of billions to trillions over time [92]. These gains do not account for qualitative gains in human well-being or in overall community level justice. Sustainable improvements must address aspects of exposure risk, the natural and built environments, government and social change [93]. More so, they must be conducive to an accessible and shared humanity.

### 5.2. Community Science

Though baseline responses from youth in this project were somewhat high overall, likely due to self-selection bias wherein already motivated youth had a higher likelihood of seeking out and enrolling in the program, improvements were apparent. The successful completion of major scientific grade tasks by youth in this and several of CERCH’s prior YPAR studies solidifies the assertion that community members and youth specifically should be meaningfully integrated into all feasible aspects of study design, implementation, analysis, and dissemination of results.

Though CERCH staff supervised all work extensively, their primary role became technical consultant rather than relying on a top-down chain of command. This approach is reflective of a commitment to fostering flat hierarchies and inclusivity across study activities. In early stages, youth continually turned to staff for advice, who in turn facilitated Freirean dialogue within the group, with the aim of helping the youth better harness their own collective assets and build a community of practice. As sampling progressed, youth increasingly consulted with each other, drawing on their “experiential expertise” to execute the majority of tasks without staff assistance. In longer term projects, this may make research processes more efficient, reducing the workloads of senior researchers and creating new opportunities for community members to build skills and meaningful opportunities for promotion.

Central tenets of YPAR draw from emancipatory pedagogy, seeking to dismantle systems of structural violence discussed and aimed at dissipating hierarchy between researchers with advanced credentials and members of communities unable to access those opportunities, outlined by Freire’s Critical Pedagogy [94]. Integrating “experiential expertise” of those with lived experiences into scientific research study design and protocols mirrors Black feminist Standpoint Theory [95], where personal experiences illuminate larger societal patterns of violence and resistance. Combined, aspects of CBPR resonate with the concept of Strong Objectivity [96], that scientific fields require increased, and innately subjective, reflection and diverse inclusion in order to acknowledge, address, and control for, human researcher’s implicit biases, thus improving aggregate objectivity. Greater diversity fosters a broader variety of subjectivities that can then cancel out, acknowledge, or balance biases implicit in demographics currently *overrepresented* in STEM fields. Each line of thought emphasizes the underutilization of diverse “everyday” knowledge or “experiential expertise”, asserting multiple ways of knowing, with academic knowledge being just one among many others yet also potentially benefiting from each.

Voices often excluded from decision-making processes are incorporated with CBPR. Community-researcher collaborations foster constructive conditions that extend far beyond data, expanding the perspectives of researchers as they conduct subsequent work and create long term opportunities for communities to access and *adapt* rigorous research methods, technologies and institutions. YPAR can improve local capacity to conduct research, better garner public interest in sciences and create desirable local employment or educational opportunities as well as provide an enriching venue for youth to engage their local community, spend time constructively, socialize with other youth and build team-work skills. 

CERCH takes special measures to support youth in college preparation such as panels with youth of color enrolled in top universities, admissions counselor meetings, personal statement workshops and deadline reminders. The work herein may also be a boon to youth’s resumes as they apply for said positions. Thus, YPAR can create pathways to college success, which may help diversify public health program enrollment and workforce. Diversification of STEM adds a multitude of perspectives and approaches, increasing the probability that new and effective interventions will be developed. Inclusion of marginalized folks may lend a more nuanced understandings of how structural violence and social determinants of health function, as these populations have familiarity with their impacts. Fostering marginalized group’s attainment of educational and employment opportunities remediates factors associated with increased chances of bearing environmental burdens. Thus, diversification is both a means to better data and interventions, and end in itself, reducing the probability of structural harm and improving the probability of developmental success. Several youth involved in this study actively continue community outreach efforts, and are pursuing or have already engaged in STEM careers. Several have also worked to foster youth mentorship, college preparedness and diversifying STEM fields for the next generation.

Short study duration and limited funding were impediments to meaningful and long term community action in the study outlined here. Drawing upon the precedent set by this study, a new grant was submitted with more than double the budget and duration. This grant includes twice the formally trained research personnel and supports a cohort of youth researchers twice as large. The new study focuses on real time measurements of indoor air pollution in relation to gas stove use in residences that include a child with asthma. This more targeted focus is more directly amenable to individual intervention and may have more immediate and observable impacts than the previous outdoor study overviewed here. The study focus is also complimentary to specific regional regulations regarding indoor air pollution that are in development, including gas stoves. An iterative, and community informed, learning process has led to a new project wherein potential policy implications are more cohesive and streamlined, a challenge faced by the study outlined in this paper.

Renewed interest in addressing structural racism, catalyzed by the murder of George Floyd, has also prompted the City of Richmond to deepen their commitment to racial justice. Much of the current focus regards police violence and incarceration inequities. We hope that our research findings, framed in the context of structural violence and indicating patterns wherein low income and people of color bear disproportionate air quality burdens, coupled with the exacerbated impacts of COVID-19, may help illuminate larger policies and systems of resource distribution that center marginalized communities, and a “freedom to breathe”.

### 5.3. Limitations of Environmental Monitoring Data

Despite our community science findings, several limitations are notable. First, our air sampling took place during a three-week period starting 21 December 2017 and ending in mid-January 2018. Therefore, our findings are generalizable only to the time period sampled and do not capture seasonal variations of air pollutant levels. That being said, we note that NO_2_ concentrations measured with the Ogawa samplers were very similar when compared against NO_2_ measurements made at the nearest US EPA ambient air quality-monitoring site (see Table 1). This finding gives added external validity to our NO_2_ measurements. We are less confident about the external validity for SO_2_ since SO_2_ levels measured were at least an order of magnitude lower than the nearest US EPA air monitoring site. We note that the two Ogawa samplers in closest proximity to the US EPA monitoring site were at least at the same order of magnitude of SO_2_ concentrations (0.12 ppb versus 0.43 ppb). However, SO_2_ observations made in our study must be interpreted with caution. Further, due to lack of resources, noise measurements were made using spot checks only and do not represent 24 h measurements, as is typically done. However, our spot measurements indicated rather high levels of ambient noise at each site location. These preliminary findings warrant further investigation and follow up.

## 6. Conclusions

Made overt by mass social movements such as Black Lives Matter, communities continue to resist at the crux of structural racism and environmental injustice. By further naming, contextualizing, and quantifying deleterious patterns of environmental disparities, these phenomena are more clearly revealed to be artificial, abnormal, and requiring redress. It is our hope that environmental health disparities, and broader awareness of intersectional structural violence, may be increasingly incorporated into social movements, as well as policy decisions. The imperative nature of issues discussed is exacerbated by the devastation COVID-19 has had, and continues to have, especially on communities of color, and especially among communities exposed to air pollution. In this regard, we encourage researchers to expand their commitment to community based participatory research and contextualize findings within historical and contemporary violence. Participatory models help generate more granular data and more efficiently render meaningful results that are actionable for multidimensional interventions, informed by deeper understandings of how structural violence, and eco-apartheid interact to probabilistically determine built and social environments that deeply affect community member’s health. Through YPAR, youth bring a fresh, often marginalized, perspective and hope, helping researchers and communities reify *why* efforts for improved health equity are of paramount importance, beyond data and abstraction. If properly conducted, the practice of community inclusion helps erode unbalanced power dynamics and socio-economic inequities, improves the long-term sustainability of actions within affected communities, and bolsters capacity to push for a healthier future. Exceeding public health as it is often bound, this may be more conducive to a collective belonging and public *healing*.

## Figures and Tables

**Figure 1 ijerph-18-00554-f001:**
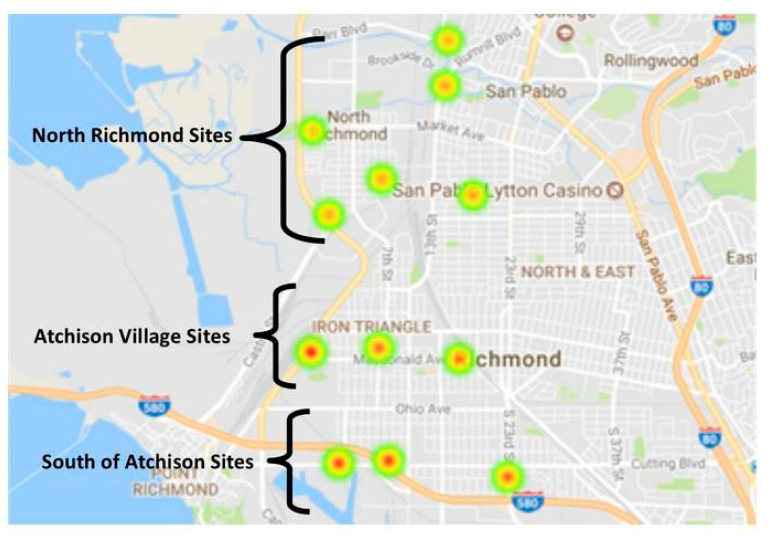
Map of sample sites for Atchison Village, South of Atchison (along I-580), and North Richmond. Darker red colors of sample sites indicate higher NO_2_ concentrations.

**Figure 2 ijerph-18-00554-f002:**
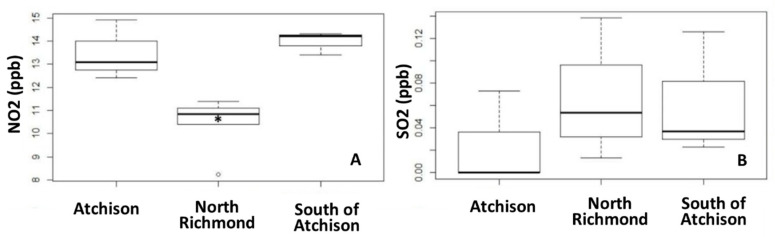
Boxplots displaying measured concentrations. (**A**) NO_2_ and (**B**) SO_2_ for each Richmond neighborhood monitored in our study (21 December–15 January 2018). Asterisk (*) indicates that North Richmond had significantly lower (*p* < 0.05) NO_2_ concentrations compared to all other monitored neighborhoods. Unfilled circles indicate outliers.

**Figure 3 ijerph-18-00554-f003:**
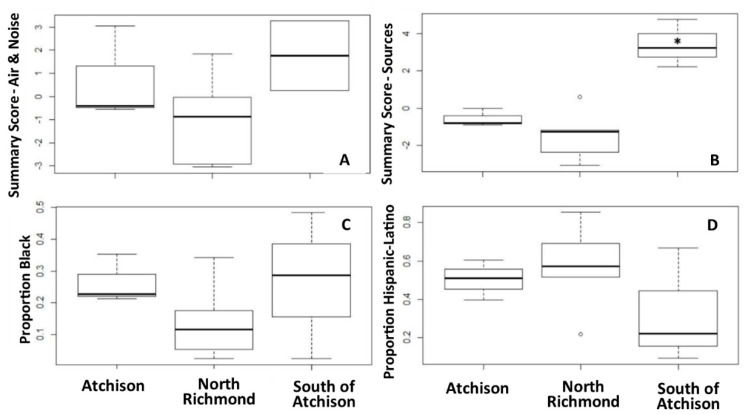
Boxplots of air and noise pollution, sources and demographics. (**A**) Neighborhood-level summary scores for air and noise pollution measurements (sum of NO_2_ + SO_2_ + noise Z-scores); (**B**) neighborhood-level summary scores for air pollution sources (sum of major roadway density + railway density + regulated site density Z-scores), Asterisk (*) indicates that North Richmond had significantly lower (*p* < 0.05) NO_2_ concentrations compared to all other monitored neighborhoods; (**C**) proportion of census tract that are Non-Hispanic Black; and (**D**) proportion in census tract that are Hispanic. Unfilled circles indicate outliers.

**Table 1 ijerph-18-00554-t001:** Average air pollution and noise pollution measurement results.

	NO_2_	SO_2_	Noise (Decibels)
Average (standard deviation)	12.09 (1.97)	0.06 (0.05)	74.64 (5.34)

Average air pollution measurements for NO_2_ and SO_2_ and decibel levels across all 12 sample sites in Richmond, CA. During the same time period, the averages from the nearest US EPA monitor is 13.24 ppb for NO_2_ and for SO_2_ is 0.430 ppb.

**Table 2 ijerph-18-00554-t002:** Validated empowerment measures with strongest pre-post improvement.

Prompt (Some Abbreviated)	Baseline	Post-Test
I can make a “real difference” in improving my city	40% strongly agreed	100%
I can “use research results to come up with realistic recommendations.”	40% strongly agreed	100%
“If I want to improve a problem in my city, I know how to gather useful data about the issue.”	20% strongly agreed	80%
“I am inclined to feel like a failure.”	20% strongly disagreed	80%
“I am satisfied with myself.”	20% strongly agreed	60%
“I want to have as much say as possible in making decisions in my school.”	60% strongly agreed	100%
“My neighborhood has lots of things I can use to help make my neighborhood better.”	0% strongly agreed	60%
“I like my neighborhood.”	20% strongly agreed	100%
“I try to help people in my neighborhood when they are in need.”	0% strongly agreed	60%
“If issues come up that affect students at my school, we do something about it.”	40% strongly agreed	80%
“If issues come up that affect youth in my city, we do something about it.”	0% strongly agreed	80%

## Data Availability

Non-human subjects raw data is included in this paper as a Appendix A.

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
