# Peer review of "“Freedom to Breathe”: Youth Participatory Action Research (YPAR) to Investigate Air Pollution Inequities in Richmond, CA"

_ijerph, 2021, doi:10.3390/ijerph18020554_

Round 1
Reviewer 1 Report
I thoroughly enjoyed reading this excellent and well-written paper, fully support the commitment to meaningful community engagement and YPAR, and look forward to being able to cite it – especially your YPAR results in section 4.2
My only criticism is that your introduction/background paragraphs could have been strengthened with the inclusion of more supporting references – you have done this best in sections 2.2.3 and 2.2.4 – can you include more references in the other paragraphs?
Also noted just the one typo on line 178 – “Because primary goal of regulatory monitoring…” should read Because the primary (or possibly Because a primary…?)
I would also just query whether the other 3 young people involved would prefer to be named rather than acknowledged as “Richmond Youth Council members” – if yes, I fully support and would recommend that IJERPH allow space for this.
Author Response
The authors thank the reviewers for their thoughtful review of this paper. Changes requested were made using track changes and are outlined below:
- Noted by Reviewer 1, the names of additional youth researchers were added to the acknowledgments, with their approval.
- Requested by Reviewer 1, additional references were added to the Introduction/Background sections, and throughout.
- Reviewer 2 requested additional details on how sampling sites were selected (lines 247-49) and this section was expanded to address this.
- Requested by Reviewer 2, examples of relevant “environmental health challenges” were added.
- Requested by Reviewer 2, additional details, comparisons and limitations were added to the Discussion section under the subheading Limitations of Environmental Monitoring Data.
- Reviewer 2 requested more details with regards to the statement “communities continue to resist”. Additional clarification was added in the Conclusion section and a new section, 2.4. Contemporary Resistance to Symptoms of Structural Violence, was added.
- Reviewer 3 requested study goals were made more explicit in the introduction. Study goals are now more clearly delineated in that section.
- Reviewer 3 made the apt points that “what you’re studying isn’t always clear” and that the discussion seemingly contained “extraneous information”. Great thought was given to these statements and improved cohesion was attempted. It is believed that the inclusion of contextual details constitute an extension of community education and advocacy, a study goal made more clear with a revision to the Abstract and the end of the Introduction section, with context and implications better articulated in the Discussion and Conclusion.
- Reviewer 3 requested details on “what next”. New information about a subsequent YPAR grant, policy implications and activities were added to the Discussion section. Some of those activities aim to build upon current social movements, and the disproportionate impacts of COVID-19, which were outlined in section 2.2.3. Contemporary Structural Violence, accordingly.
Reviewer 2 Report
This manuscript covers a project aimed to facilitate empowerment among high school youth thru environmental health literacy, professional development; and participation in designing and conducting research on air pollution and community-level health risk factors.
Researchers measured ambient air pollution in a low-income community near industry-emitting sources on Richmond, CA. This community currently has one regulatory air monitoring site to measure criteria air pollutants so little is known about spatial variability in exposure within this community so youth sought to characterize exposure to ambient air pollution and noise.
The manuscript also provides an in-depth overview of structural violence and how it trickles into environmental justice issues in marginalized communities. The study implemented the YPAR approach, a corollary of CBPR with a focus on youth empowerment and trajectories development.
Overall, the paper is well written and offers a novel perspective into how to begin addressing structural environmental health racism/violence in marginalized communities.
This reviewer recommends accepting the manuscript after all reviewer comments are addressed including the ones noted below.
- Line 28-35: The sentences included sound like instructions to authors. Please delete
- Line 247-249: Please expand on how air monitoring sites were selected within the burdened communities using the tools indicated.
- Line 192: Please expand on “environmental health challenges.”
- Discussion: Authors need to include a section that compares their environmental measurements (air pollution and noise) to other studies to put the results into context. Additionally, they did not address the limitations of their environmental monitoring (e.g., variability by season, how the levels compared to the actual EPA monitoring site, etc).
- Conclusion: Please expand on “communities continue to resist.” Resist what? Focus the sentence and clarify for readers.
Author Response
The authors thank the reviewers for their thoughtful review of this paper. Changes requested were made using track changes and are outlined below:
- Reviewer 2 requested additional details on how sampling sites were selected (lines 247-49) and this section was expanded to address this.
- Requested by Reviewer 2, examples of relevant “environmental health challenges” were added.
- Requested by Reviewer 2, additional details, comparisons and limitations were added to the Discussion section under the subheading Limitations of Environmental Monitoring Data.
- Reviewer 2 requested more details with regards to the statement “communities continue to resist”. Additional clarification was added in the Conclusion section and a new section, 2.4. Contemporary Resistance to Symptoms of Structural Violence, was added.
Reviewer 3 Report
I appreciated the opportunity to review your paper. Participatory methods such as those you used are extremely relevant and stand to produce both a process and outcome that is beneficial to communities most impacted by the issues at hand. Overall I think this is a strong paper, well-written, and meticulous. However, I do have some comments that I believe could strengthen the paper overall.
Most of my comments center on the challenge of deciphering the methods from the results. The "what" you're studying isn't always clear because it seems to be both substantive content and the process/outcomes of YPAR. I think this is a common challenge when presenting results from CBPR studies, and can be rectified by making clear statements that delineate these things and explicitly discuss how they overlap in the methods section.
In the introduction, you discuss project goals, but you do not discuss the present study's goals. I think it would be better if you shifted lines 272-277 (or repackaged them) to the introduction. It would be helpful for the reader to have the explicit study purpose in mind before reading background.
In the discussion, there seems to be extraneous information (e.g. about the project and its impacts that are peripheral to the present study). Suggest removing those and keeping the discussion more tightly focused around the study purpose.
One thing that is a real hallmark of participatory methods is the "what next." You presented the growth among youth that occurred, but what does this set them and their communities up for next? In your discussion you note that YPAR helps generate more granular data and produces more meaningful results that are actionable for multidimensional interventions. Can you speak to the "what next" in terms of interventions for this particular community, especially as they relate to the substantive findings?
Author Response
The authors thank the reviewers for their thoughtful review of this paper. Changes requested were made using track changes and are outlined below:
- Reviewer 3 requested study goals were made more explicit in the introduction. Study goals are now more clearly delineated in that section.
- Reviewer 3 made the apt points that “what you’re studying isn’t always clear” and that the discussion seemingly contained “extraneous information”. Great thought was given to these statements and improved cohesion was attempted. It is believed that the inclusion of contextual details constitute an extension of community education and advocacy, a study goal made more clear with a revision to the Abstract and the end of the Introduction section, with context and implications better articulated in the Discussion and Conclusion.
- Reviewer 3 requested details on “what next”. New information about a subsequent YPAR grant, policy implications and activities were added to the Discussion section. Some of those activities aim to build upon current social movements, and the disproportionate impacts of COVID-19, which were outlined in section 2.2.3. Contemporary Structural Violence, accordingly.